# The Effect of N-Carbamylglutamate Supplementation during the Last Third of Gestation on the Growth and Development of Fetuses Born to Nutrient-Restricted Twin-Bearing Ewes

**DOI:** 10.3390/ani14060946

**Published:** 2024-03-19

**Authors:** Víctor H. Parraguez, Susan McCoard, Camila Sandoval, Francisca Candia, Paul Maclean, Wade Mace, Xinqi Liu, Francisco Sales

**Affiliations:** 1Faculty of Veterinary Sciences, University of Chile, Santiago 8820808, Chile; vparragu@uchile.cl (V.H.P.); francisca.candia.o@ug.uchile.cl (F.C.); 2Faculty of Agrarian Sciences, University of Chile, Santiago 8820808, Chile; 3AgResearch Grasslands, Private Bag 11008, Palmerston North 4442, New Zealand; sue.mccoard@agresearch.co.nz (S.M.); paul.maclean@agresearch.co.nz (P.M.); wade.mace@agresearch.co.nz (W.M.); leo.liu@agresearch.co.nz (X.L.); 4INIA Kampenaike, Punta Arenas 6212707, Chile; camila.sandoval.torres@inia.cl

**Keywords:** ovine gestation, fetal growth, N-carbamylglutamate, nutrient restriction, lamb survival

## Abstract

**Simple Summary:**

Improvement of fetal growth in twin pregnancies is one of the major goals of extensive sheep production systems, as lambs’ birth weight is correlated with their survival. This is especially relevant in harsh environments where natural maternal nutrient restriction is common. Maternal arginine (Arg) supplementation improves fetal growth, but Arg is degraded in the rumen, has a short biological half-life, and the high cost of protected forms limits its use in extensive sheep systems. N-carbamylglutamate (NCG) is not degraded in the rumen, increases arginine synthesis, and improves urea recycling, suggesting it may be an option as an alternative to Arg. The aim of this study was to evaluate the effect of oral NCG supplementation from 100 days of gestation (dga) to term in naturally nutrient-restricted grazing twin-bearing ewes, on maternal, placental, and fetal variables. The serum concentrations of NCG were increased 15-fold, and the plasma urea, albumin and phosphorus concentrations were reduced in supplemented ewes compared to controls, but there was no major effect on the dam or fetal body or organ weights nor the antioxidant markers and other blood biochemical parameters. These results indicate that NCG supplementation in mid-to-late gestation in grazing ewes was unable to rescue the negative production effects of severe natural nutritional restriction on both the dam and fetuses. The challenge of meeting the nutrient requirements of pregnant ewes in harsh environmental conditions reinforces the need for future research to identify novel strategies to improve lamb survival in such environments around the world.

**Abstract:**

N-carbamylglutamate (NCG) is postulated to improve fetal growth in nutrient-restricted gestations when supplemented from day 35 to 110 of gestation, but the effects of supplementation from 100 days of gestation to birth have not been evaluated. The aim of this study was to evaluate the effect of oral NCG supplementation from 100 days of gestation (dga) to term in naturally nutrient-restricted grazing twin-bearing ewes, on the maternal body weight (BW), body condition score (BCS), placental morphology, fetal body and organ weights and blood biochemistry and antioxidant status in the ewe and fetuses. Eighteen twin-bearing ewes maintained under grazing management were randomly allocated to either a treatment group (NCG; *n* = 10), orally dosed once daily with 60 mg/kg of NCG from day 100 until 140 dga, or an unsupplemented control group (CON; *n* = 8). At 140 dga, blood gases, redox status, maternal and fetal plasma and fetal biometrics were obtained after caesarian section. The serum concentration of NCG was increased 15-fold in the NCG ewes compared to the CON. No major effects on dam or fetal body weight nor on blood biochemistry or antioxidant parameters were observed. These results indicate that NCG supplementation in mid-to-late gestation to grazing ewes was unable to rescue the negative production effects of severe natural nutritional restriction on both the dam and fetuses.

## 1. Introduction

Lamb survival in the first few days of life, especially in multiple gestations, is one of the most common factors limiting sheep productivity worldwide [1]. Lamb survival is a multifactorial trait [2,3]. Maternal nutrient restriction during gestation leads to fetal growth restriction and low lamb birth weight and is a key contributor to lamb mortality [4,5]. Several targeted interventions have been evaluated to explore options to reduce the negative effect of maternal nutrient restriction during pregnancy and improve the lifetime performance of the offspring [6]. However, strategies during last third of gestation to improve lamb survival around birth in nutrient-restricted ewes remain elusive.

L-Arginine (Arg) is a versatile amino acid [7], which has been shown to improve not only fetal growth when supplemented to nutrient-restricted ewes [8] but also traits important for lamb survival in multiple gestations [9,10]. However, Arg is degraded in the rumen [11], has a short biological half-life [12], and the reduced availability and high cost of protected forms limits its use in extensive sheep systems [13].

N-carbamylglutamate (NCG), an analogue of N-acetylglutamate (a key enzyme in the synthesis of Arg), is postulated to improve Arg synthesis, urea recycling, nitrogen utilization and antioxidant capacity in ruminants [14]. Interestingly, NCG is not extensively degraded in the rumen [11] and the cost of NCG compared to Arg is significantly lower [15]. Supplementation of NCG to nutrient-restricted ewes (50% NRC) before 110 days of gestation ameliorates fetal growth restriction in multiple-gestation sheep [16,17], improving placental growth, antioxidant capacity in maternal and fetal plasma [18], placental [19] and intestinal [20] amino acid transport, fetal thymus development and immune function [21] when evaluated at 110 days of gestation. However, the impact on fetal growth and development after 110 days of gestation in sheep has not been reported. One study in dairy cows suggests that maternal supplementation with NCG during the last 28 days of pregnancy improved the newborn weight via improvement of angiogenesis, nutrient transport and urea cycle in the fetus [22]. It is not clear if a similar time of intervention in pregnant sheep would improve the fetal outcome. 

Nutrient restriction in sheep during pregnancy is common in Patagonia, with negative effects on ewes’ body condition score (BCS) and reproductive performance [23]. This is critical in multiple gestations, as twin-bearing ewes have increased nutrient requirements compared to singletons [24]. The nutrient requirements of twin-bearing ewes increase relative to singletons in the last third of gestation and interventions during this timeframe have the potential to improve intrauterine and fetal growth and development in twin pregnancies [25]. 

The aim of this study was to evaluate the effect of oral NCG supplementation from 100 days to term in naturally nutrient-restricted grazing twin-bearing ewes, on the maternal body weight (BW) and BCS, placental morphology, fetal weight and body composition and maternal and fetal blood biochemical and antioxidant parameters. We hypothesized that maternal NCG supplementation to naturally nutrient-restricted twin-bearing ewes from 100 to 140 days of gestation would enhance the growth, metabolic status and antioxidant capacity of the fetus compared to unsupplemented controls.

## 2. Materials and Methods

### 2.1. Ethics Statement

This study was performed in agreement with the *Guide for Care and Use of Laboratory Animals* (Eighth Edition, National Research Council, National Institute of Health, USA). The protocol was approved by the Bioethics Review Committee of the Instituto de Investigaciones Agropecuarias (INIA, Ministry of Agriculture, protocol N° 07-2022).

### 2.2. Animals and Experimental Procedure

This study was conducted at INIA-Kampenaike Research Farm, 60 km north of Punta Arenas, Chile (Chilean Patagonia, Lat. 52°41′ S Long. 70°54′ W). Corriedale ewes (4–6 years old) from a commercial flock were synchronized using an intravaginal progesterone device (CIDR G^®^, Pfizer, Santiago, Chile) for 12 days, followed by 300 IU equine chorionic gonadotrophin (eCG; Novormon, Syntex, Argentine) via i.m. injection. Mating was carried out via laparoscopic artificial insemination, using a single Corriedale ram to standardize the paternal effect. Transabdominal ultrasound pregnancy examination was performed 80 days after mating (Oviscan, BCF Technology Ltd, Livingston EH54 8TE, Scotland, UK), and 18 twin-bearing ewes were selected. The weight and body condition scores (BCSs, 1 to 5 scale) were similar for all the animals (57.98 ± 2.34 kg and 2.7 ± 0.36, respectively) at 100 days of gestation (dga). Ewes were randomly assigned into two groups. The first group, defined as NCG (*n* = 10), was orally dosed once daily with 60 mg/kg BW NCG (Inner Mongolia Tianyi Sci. & Tech. Co., Ltd., Ulanqab, Inner Mongolia, China) in a water carrier from 100 to 140 dga. Each NCG dose was prepared according to the individual ewe’s weight. The control group received no supplementation (CON; *n* = 8). All the animals received the same daily management and handling to reduce the impact of handling stress on the treatment response. The ewes were managed together as a single mob in a paddock with natural pasture (4.23% CP, 1.69 Mcal/kg ME), with a stocking rate of 0.9 ewes per hectare and a dry matter availability of 255 kg per hectare, representative of commercial Patagonian prairie conditions. Due to the severe nutrient restriction resulting from environmental conditions, all the ewes were supplemented with alfalfa hay (600 g/day/ewe; 13% CP and ME 2.2 Mcal/kg ME, on a dry matter basis) from 130 to 140 dga. The maternal BW and BCS were recorded every 10 days, from 100 until 140 dga.

At 140 dga, maternal and fetal blood samples were collected following caesarian section, as previously described [25]. Maternal jugular (10 mL) and umbilical cord (5 mL) venous blood samples were collected for evaluation of the blood chemistry and oxidative status. The weight of each fetus, sex, crown–rump length (CRL), front and hind leg length, thoracic girth circumference and weights of the internal organs were recorded. Placentas were extracted and weighted. Each placentome was dissected from the maternal and fetal membranes and weighed individually for the assessment of the total placentome number, total placentome weight, mean placentome weight and placental efficiency, estimated as the ratio of total litter weight (g) per total placental weight (g). 

### 2.3. Assessment of Oral NCG Flux into Maternal Circulation

The concentration of NCG in the maternal plasma was determined via liquid chromatography triple quadrupole mass spectroscopy (LC-QQQ-MS). Samples (100 µL) were first diluted with 500 µL of 50% methanol before briefly vortexing to mix, and then centrifuging (14,000 rcf for 10 min at 4 °C) using a Centrifuge 5427 R (Eppendorf, Hamburg, Germany). A 500 µL aliquot of the diluted sample was added to a pre-conditioned (1 mL methanol followed by 1 mL of deionized water) reversed-phase anion-exchange solid-phase extraction cartridge (Strata-X-A, 33 µm, 200 mg/3 mL, Phenomenex, Torrance, CA, USA). The cartridge was rinsed with 1 mL each of 25 mM ammonium acetate solution then methanol. NCG was eluted using 2 × 1 mL of 5% formic acid in methanol. The eluate was dried using a CentriVap Concentrator attached to a FreeZone 4.5 L (−105 °C) freeze-drier (Labconco Corporation, Kansa City, MO, USA) before being reconstituted in 100 µL of methanol and transferred to amber HPLC vials for analysis.

NCG was detected and quantified using an LCMS-8060NX (Shimadzu Corporation, Kyoto, Japan). Separation of the NCG was achieved using a Poroshell HILIC-Z 150 × 2.1 mm (2.7 µm) column (Agilent Technologies, Santa Clara, CA, USA) eluted at a flow rate of 400 µL/min with 16 mM ammonium formate (A) and 97% acetonitrile containing 0.1% formic acid (B) using the following gradient: Start at 92.8% B, T 10 min 64.8% B, T 11.5 min 64.8% B, and T 12 min to initial conditions with a 3 min equilibration. NCG was detected and quantified using the 189–146 *m*/*z* mass fragmentation transition with the 186–128 *m*/*z* mass transition used for compound confirmation. The analyte peaks were detected and quantified using LabSolutions Insight v4.0 (Shimadzu Corporation, Kyoto, Japan). The limits of detection and quantitation using this method were 2 ng/mL and 6 ng/mL, respectively.

### 2.4. Assessment Umbilical Blood Gases, Chemistries, Hematocrit and Hemoglobin

At 140 dga, 1 mL of blood was obtained from the umbilical vein of all the fetal sheep into heparinized syringes for blood gas analysis. The analysis was performed immediately after blood collection with a portable blood gas analyzer (I-STAT^®^, Abbott Laboratories, Abbott Park, IL, USA) and the CG8+ cartridge (Abbott Lab., Abbott Park, IL, USA), as previously described for ewes [26]. The partial pressure of oxygen (PO_2_), partial pressure of carbon dioxide (PCO_2_), total carbon dioxide (TCO_2_), hemoglobin concentration (Hb), bicarbonate (HCO_3_), oxygen saturation (sO_2_), base excess (BE), sodium (Na), potassium (K), calcium (Ca), hematocrit (Hct), and pH value were measured using this approach. 

### 2.5. Assessment of Oxidative Stress Biomarkers

The maternal and fetal plasma redox statuses were evaluated at 140 dga by measurement of the malondialdehyde (MDA) concentration and total antioxidant capacity (TAC), as previously described [23].

### 2.6. Assessment of Maternal Plasma Metabolites

Maternal blood samples (10 mL) were obtained every 10 days from 100 to 140 dga. Blood was collected from the jugular vein and transferred to plain vacuum tubes (BD Vacutainer, Franklin Lakes, NJ, USA). The tubes were centrifuged at 4 °C and 3000× *g* for 10 min; serum was stored at −20 °C until assayed. The determined metabolites, methods and laboratory information is presented in Table 1.

### 2.7. Statistical Analysis

All the data were assessed for normality and homogeneity and no transformation was necessary. The effects of the treatment on the ewes’ BW, BCS and plasma metabolites were analyzed using REML repeated measures analysis in R using the lme4 R package (R version 4.3.0.) with the time (dga) and treatment (and their interaction) as fixed effects and the ewe as the random effect. The serum NCG, feto-placental blood gases, chemistries, hematocrit and hemoglobin, oxidative stress biomarkers and placental traits were analyzed using REML analysis in R using the lme4 R package with the treatment as the fixed effect and the ewe as the random effect. The fetal weight, measurements and organ weights were evaluated considering the treatment, sex of the fetus and the interaction as fixed effects, with the ewe as the random effect, to account for twin pregnancies. When presented, all the post hoc tests were performed using Fisher’s Least Significant Difference (LSD) test. A treatment difference was considered significant if *p* < 0.05 and as a trend when *p* < 0.10.

## 3. Results

### 3.1. Maternal Serum NCG

The maternal serum concentrations of NCG were 15-fold higher in the treated ewes relative to the controls 14 h post-dosing at 140 dga (290.1 ± 16.86 vs. 19.3 ± 16.86 ng/mL, *p* < 0.001). Low levels of endogenous serum NCG are normally detected.

### 3.2. Maternal Body Weight and BCS

The ewes’ BW increased (*p* < 0.0001) and BCS decreased (*p* = 0.005) with advancing gestation, but there was no effect of treatment (Figure 1). When the conceptus weight was discounted from the final maternal BW to estimate only the maternal BW change in the period, a ~17% BW loss was observed, with no difference between the NCG and CON animals (48.03 ± 0.92 vs. 48.69 ± 1.10 kg, respectively; *p* = 0.65).

### 3.3. Fetal Body Measurements

The effect of maternal oral supplementation with NCG compared to controls on the fetal weight, body dimensions and organ weight is presented in Table 2. No effect of treatment (*p* > 0.05) was observed for the fetal body weight and dimensions. Fetuses from the NCG ewes had 22% lower lung weights compared to their control counterparts (*p* = 0.023). There was a tendency for a 5% lower brain weight in the NCG compared to the CON fetuses (*p* = 0.07). Overall, male fetuses had 10% heavier intestine weights (*p* = 0.03) and tended to have 10% heavier stomachs (*p* = 0.08) compared to females. No other effects of treatment on the fetal organ weights were observed.

Table 3 presents the effects of treatment, sex and their interaction on the fetal body weight and composition data with the fetal weight as a covariate. There was a tendency for a treatment by fetal sex interaction for the CRL (*p* = 0.09), heart (*p* = 0.08) and right kidney weight (*p* = 0.07), where male fetuses from CON ewes had 3% greater CRL, 11% greater heart and 20% greater right kidney weight compared to female fetuses from CON ewes, while no differences between the sexes was observed for fetuses from NCG ewes. There was a trend for fetuses from NCG ewes to have disproportionately lower brain (*p* = 0.08) and lung weights (*p* = 0.07) and disproportionately larger small and large intestine weight (*p* = 0.07) compared to fetuses from CON ewes. 

### 3.4. Umbilical Blood Gases, Chemistries, Hematocrit and Hemoglobin

The effects of treatment, sex and their interaction on the blood parameters are presented in Table 4. A treatment by sex of the fetal lamb interaction was observed for sodium, calcium, pH, and PCO_2_, where no differences were observed between the sexes in the fetuses from NCG ewes, while males from CON ewes had higher calcium and PCO_2_ but lower pH concentrations compared to females. Independent of treatment, male fetuses had higher serum concentrations of potassium and lower BE compared to females. No additional effects were observed for the other blood gases (PO_2_, TCO, sO_2_, HCO_3_, BE, electrolytes (Na and K), hematocrit and hemoglobin (*p* > 0.05).

### 3.5. Oxidative Stress Biomarkers

No treatment effects were observed for the maternal or fetal TAC (Figure 2A) or MDA (Figure 2B; *p* > 0.05).

### 3.6. Maternal Plasma Metabolites

Repeated measures analysis showed a treatment by time interaction for plasma phosphorus (*p* = 0.02), where a divergence in their levels was evident between 130 and 140 dga, resulting in ~10% lower concentrations (*p* = 0.003) in the NCG relative to the CON ewes by 140 dga. A similar trend was observed for plasma albumin (*p* = 0.09), with a divergence between groups between 130 and 140 dga, resulting in ~6% lower plasma albumin (*p* = 0.002) in the NCG compared to the CON ewes at 140 dga (Figure 3).

The plasma metabolites from the NCG-supplemented and CON ewes at 140 dga are presented in Table 5. For the serum concentration of urea, a trend for a treatment effect was observed at 140 dga, where the NCG ewes tended to have~10% lower concentrations compared to the CON ewes (*p* = 0.07). No effects of treatment on the other blood metabolites at 140 dga were observed. 

### 3.7. Placental Traits

There was no difference in the total placental weight, total placentome weight and number, mean placentome weight or placental efficiency between the NCG-supplemented and CON ewes (Table 6).

## 4. Discussion

The key finding of this study was that while the serum concentrations of NCG were increased 15-fold in the supplemented ewes compared to the controls, there was no major effect on the dam or fetal body weight, composition or blood antioxidant parameters and only limited effects on the blood biochemical parameters. These results indicate that NCG supplementation (60 mg/kg) in mid-to-late gestation to grazing ewes was unable to rescue the negative physiological and developmental effects of severe natural nutritional restriction on both the dam and fetuses when supplemented from 100 to 140 dga. 

Nutrient restriction normally occurs in rangeland areas, where sheep often experience bouts of nutrient restriction of less than 50% of their nutritional requirements [27]. Similar levels of nutritional restriction of pregnant ewes during winter are observed in extensive grazing systems in Patagonia. Low forage availability and quality (i.e., low CP and ME), particularly during pregnancy [28], make it difficult to meet maternal and fetal nutritional requirements, resulting in negative effects on production performance. Ewes tend to lose at least one BCS unit and over a 20% of their body weight during the early gestation period (before 100 dga) in twin-bearing ewes [23]. In the present study, both the NCG and CON ewes exhibited a similar BCS and BW loss compared to what was observed in one of our previous studies [29]; however, the negative effects of undernutrition on the dam and fetal parameters were not ameliorated with NCG supplementation. The absence of an effect of NCG on maternal weight is in agreement with Zhang et al., where twin-bearing ewes nutrient-restricted (50% NRC) were supplemented from day 35 to 110 of gestation with NCG and showed no effect on maternal weight at 110 dga [17]. Similarly, Arg supplementation to 50% nutrient-restricted ewes from day 100 to 125 [30], or from 60 to parturition [8], or in 60% nutrient restricted ewes supplemented rumen-protected arginine supplement dosed at 180 mg/kg BW during last two-thirds of gestation [31], resulted in no difference in maternal body weight, consistent with the results of the present study. 

The high level of natural nutrient restriction had a metabolic impact in both groups, reflecting the harsh environment encountered in sheep-grazing systems in Patagonia. During late gestation, an increase in NEFA is normally expected, resulting from a negative energy balance and fat mobilization [32]. However, this increase in more marked when ewes are under-fed. For example, twin-bearing ewes restricted at 30% of their nutrient requirements (CP and ME), during 15 days starting at 100 dga, showed a 200% fold increase in circulating NEFA [33]. In the present study, the plasma NEFA concentrations observed in both groups exceeded the normal range for sheep (0–300 µmol/L) and the cutoff point for developing clinical adverse metabolic conditions (390 µmol/L) [34], indicating severe nutrient restriction and negative energy balance in both groups of animals. The low nutrient supply was supported by the pasture nutrient composition analysis. The lower albumin and urea concentrations in the NCG relative to the CON ewes is consistent with the potential for increased urea recycling in response to NCG supplementation [14]; however, this was unable to rescue the negative effects of severe maternal nutritional restriction on the maternal or fetal production parameters. It is important to note that the divergence in albumin and urea concentrations between the groups was evident between 130 and 140 dga, corresponding with the period of alfalfa supplementation. These results suggest that the effect of NCG could be related to the maternal nutrient level. Further studies are required to evaluate the interaction between the level of maternal nutritional restriction and NCG supplementation on urea recycling and other metabolic changes. 

No effect on the fetal weight, placental weight, morphology or efficiency was observed, suggesting that NCG supplementation did not counteract the negative effects of maternal undernutrition on fetal growth in twin pregnancies. This result contrasts with the positive effect on fetal growth observed at 110 dga following maternal NCG supplementation to nutrient-restricted (50% NRC) twin-bearing ewes from day 35 to 110 of gestation, using an effective dose of 2.5 g of NCG per ewe (~60 mg/kg BW in 40 kg ewes). In this study, a 21% increase in fetal weight and some changes in placental parameters at 110 dga were reported [17]. However, in that study fetal weight and placental parameters later in gestation were not reported. Therefore, it is possible that supplementation earlier in pregnancy may be required to elicit an effect on placental development and fetal growth in nutrient-restricted ewes, as previously shown to occur also in goats supplemented from day 0 to 90 of gestation [35]. However, supplementation of nutritionally restricted (50% NRC) single-bearing ewes from 100 to 125 dga with Arg also failed to elicit a fetal growth or placental development effect [30]. In contrast, when animals are fed to meet or exceed nutritional requirements, NCG supplementation during the last 28 dga in cattle [22], or Arg supplementation from 100 dga to term in twin-bearing ewes [10], has been reported to have a positive effect on fetal growth. Collectively, these results suggest that the level of nutritional restriction on the ewes in the present study may have been too severe to be rescued by NCG supplementation in mid-late gestation, when the nutritional requirements are the greatest.

Consistent with the lack of effect on the fetal weight, there were minimal differences in the fetal tissue and organ weights in response to maternal NCG supplementation. In contrast to prior studies, where brown fat deposition was enhanced with maternal Arg supplementation from 100 to 140 dga [10], there was no effect observed in response to NCG supplementation in the present study. The smaller brain and lungs of fetuses from the NCG compared to the CON ewes was intriguing. The growth-restricted fetus adapts its circulation to preserve oxygen and nutrient supply to the brain (“brain-sparing”) [36], and increased brain and lung weight have been previously described at 110 dga in fetuses from nutrient-restricted ewes supplemented with NCG [17], contrasting with the results in the present study. Also, no effect of thymus growth was observed, contrasting with the previously described effect for NCG supplementation [21]. It is likely that the level of nutrient restriction in the present study exceeded the capacity for the fetus to compensate for the normal growth of these organs. In contrast, the disproportionately larger combined small and large intestine weight in the NCG compared to the CON fetuses may be an adaptive mechanism to support enhanced nutrient uptake in the early post-natal period. It is important to note that the aforementioned differences in the brain, lung and intestine weights did not reach statistical significance, and the differences were relatively small and therefore may not be biologically significant, and they would require validation in future studies. 

Twin pregnancies in nutrient-restricted gestations occur under a hypoxic environment, leading to oxidative stress and contributing to reduced fetal growth [29]. Oxidative stress results from the imbalance between cellular the natural antioxidant defenses and pro-oxidant state, augmenting the concentration of reactive oxygen species (ROS) [37]. Targeted interventions with antioxidants during the entire gestation has proved to counteract oxidative stress in naturally undernourished pregnant ewes, improving fetal growth, regardless of the liter size [25]. The NCG antioxidant capacity has been demonstrated in lambs [38,39] and dairy cows [40] and has also been proposed to improve the maternal–fetal–placental antioxidant capability in underfed twin-bearing ewes (50% NRC) at 110 dga by increasing TAC and reducing MDA in maternal and fetal plasma, and in the caruncle and cotyledon, when supplemented from day 35 to 110 of gestation [18] via modification of specific metabolic pathways [15]. However, in the present study, maternal supplementation with NCG during late gestation resulted in no effects on the materno-fetal antioxidant status. Previous studies have shown that pregnant ewes maintained in a nutrient imbalance given by low energy availability [37] or undernutrition (30% NRC) leading to a lipid imbalance [41] have decreased antioxidant capacity during late pregnancy, and the level of reduction in the antioxidant capacity is related to the severity of the nutrition insult. Therefore, some plausible explanations for the lack of response to NCG supplementation in the present study could be the timing of the supplementation during late gestation, contrasting with intervention before 110 dga [18], or the level of maternal nutritional restriction was so severe that it could not be recovered via NCG supplementation at the level used in the present study. More research would be required to study the potential benefits of NCG for oxidative stress and the interaction with the level of maternal nutritional restriction.

## 5. Conclusions

The results of the present study indicate that short-term oral maternal supplementation with NCG during mid–late gestation and under natural grazing severe nutrient restriction potentially improved urea recycling but was not able to improve maternal or fetal traits important to support lamb survival and growth. Future studies could consider evaluating the interaction between supplementation and the level of nutritional restriction, timing of the intervention (e.g., earlier intervention) and dose-response effects may be considered. The challenges of meeting the nutrient requirements of pregnant ewes in harsh environmental conditions such as Patagonia and the impacts on animal performance were also highlighted, reinforcing the need for future research to identify novel strategies to improve lamb survival in such environments around the world.

## Figures and Tables

**Figure 1 animals-14-00946-f001:**
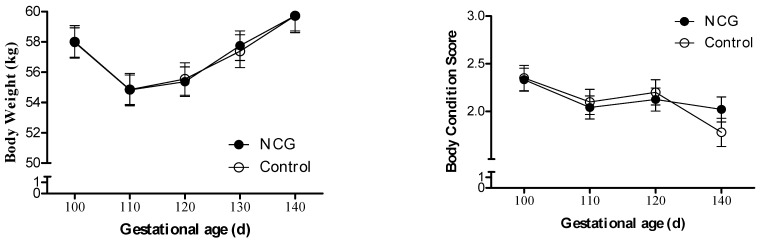
Body weight and body condition score variation from day 100 to 140 of gestation of NCG-supplemented (NCG) and unsupplemented control (CON) ewes.

**Figure 2 animals-14-00946-f002:**
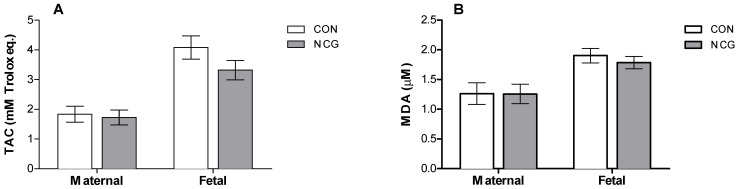
Mean (±SEM) plasma TAC (**A**) and MDA (**B**) as oxidative stress biomarkers of supplemented (NCG) and control (CON) ewes and fetuses at 140 days of gestation. TAC: total antioxidant capacity; MDA: malondialdehyde.

**Figure 3 animals-14-00946-f003:**
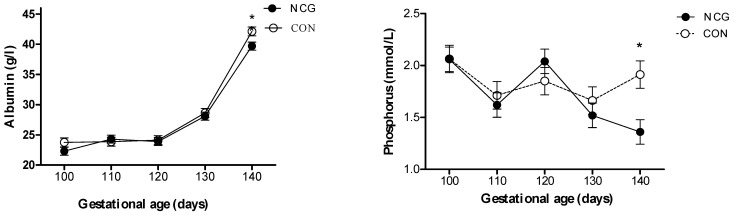
Serum concentrations of albumin (g/L) and phosphorus (mmol/L) from day 100 to 140 of gestation in supplemented (NCG) and control (CON) ewes. * *p* < 0.05 between NCG and CON.

**Table 1 animals-14-00946-t001:** Determined metabolites, methods, and laboratory information.

Metabolite	Method	Kit	Kit n°	Laboratory
Non-esterified fatty acids (NEFA)	Colorimetric method	Randox	FA 115	Randox
Urea	Enzymatic-colorimetric method	Human	10505	Wiesbaden
Total protein	Colorimetric test	Human	157004	Wiesbaden
Albumin	Colorimetric test	Human	10570	Wiesbaden
Globulin	Indirect estimation			
Calcium	Colorimetric test	Human	10011	Wiesbaden
Phosphorus	Colorimetric test	Human	10027	Wiesbaden
Magnesium	Colorimetric test	Human	10010	Wiesbaden
Cholesterol	Colorimetric enzymatic test	Human	10017	Wiesbaden
Aspartate aminotransferase (AST)	Enzymatic UV test	Human	12211	Wiesbaden
Creatine kinase (CK)	Enzymatic UV test	Human	12015	Wiesbaden
Gamma-glutamyltransferase (GGT)	Enzymatic color test	Human	12213	Wiesbaden
Glutamate dehydrogenase (GLDH)	Colorimetric	Randox	GL441	Randox
Total bilirubin	Colorimetric test	Human	10743	Wiesbaden
Triglycerides	Enzymatic colorimetric test	Human	10720P	Wiesbaden

**Table 2 animals-14-00946-t002:** Body weight, skeletal dimensions, organ and digestive tract weights of fetuses at 140 days of gestation from twin-bearing ewes supplemented with NCG compared to unsupplemented controls (CON) ^1^.

	CON		NCG		*p-*Value		
	Female	Male	Female	Male	Treat	Sex	Treat × Sex
Body weight (kg) and dimensions ^2^ (cm)							
Fetal weight	3.39 ± 0.15	3.54 ± 0.17	3.23 ± 0.11	3.38 ± 0.13	0.30	0.20	0.97
Crown–rump length	39.70 ± 0.71	41.30 ± 0.81	39.95 ± 0.53	39.87 ± 0.63	0.61	0.28	0.16
Thoracic girth	32.38 ± 0.77	32.54 ± 0.92	32.59 ± 0.58	31.76 ± 0.72	0.87	0.52	0.48
Front leg length	28.98 ± 0.95	31.06 ± 1.20	30.05 ± 0.74	29.24 ± 1.01	0.99	0.71	0.16
Hind leg length	33.62 ± 0.56	33.72 ± 0.70	33.20 ± 0.43	33.69 ± 0.57	0.61	0.55	0.74
Organ weight (g) ^2^							
Semitendinosus m	5.39 ± 0.36	5.39 ± 0.43	4.97 ± 0.27	5.28 ± 0.34	0.43	0.56	0.63
Brain	53.74 ± 1.36	55.60 ± 1.57	51.17 ± 1.01	52.25 ± 1.20	0.07	0.19	0.73
Heart	24.75 ± 1.56	28.62 ± 1.82	24.68 ± 1.15	25.29 ± 1.41	0.42	0.14	0.23
Lungs	125.21 ± 9.04	131.16 ± 11.51	101.14 ± 7.01	108.40 ± 9.60	0.02	0.48	0.95
Liver	80.96 ± 6.44	86.23 ± 7.74	71.60 ± 4.81	76.84 ± 6.10	0.17	0.36	0.99
Spleen	5.65 ± 0.54	6.85 ± 0.62	5.08 ± 0.40	5.40 ± 0.48	0.13	0.12	0.33
Right kidney	8.56 ± 0.74	10.46 ± 0.88	8.98 ± 0.55	9.30 ± 0.69	0.79	0.14	0.24
Left kidney	8.69 ± 0.79	10.28 ± 0.96	9.20 ± 0.59	9.15 ± 0.76	0.89	0.40	0.28
Perirenal fat	18.96 ± 1.54	16.48 ± 1.97	17.29 ± 1.20	15.58 ± 1.65	0.40	0.22	0.82
Thymus	13.20 ± 1.50	14.15 ± 1.80	12.38 ± 1.12	13.38 ± 1.41	0.60	0.46	0.99
Thyroid glands	0.91 ± 0.10	0.96 ± 0.11	0.98 ± 0.07	0.99 ± 0.08	0.62	0.66	0.77
Adrenal glands	0.41 ± 0.03	0.46 ± 0.03	0.39 ± 0.02	0.40 ± 0.03	0.26	0.29	0.39
Digestive tract weight (g) ^2^							
Stomach	38.13 ± 2.91	39.37 ± 3.38	33.64 ± 2.16	39.81 ± 2.60	0.38	0.08	0.32
Intestine ^3^	121.61 ± 8.11	135.04 ± 9.03	129.51 ± 6.13	141.91 ± 6.94	0.45	0.03	0.93

^1^ Data are presented as the means (predicted from the fitted statistical model) ± SEM, together with the *p* value for treatment, sex and treatment by sex interaction effects. ^2^ Data were analyzed without fetal weight as a covariate. ^3^ Small and large.

**Table 3 animals-14-00946-t003:** Body weight, skeletal dimensions, organ and digestive tract weights of fetuses, adjusted by fetal weight, at 140 days of gestation from twin-bearing ewes supplemented with NCG compared to unsupplemented controls (CON) ^1^.

	CON		NCG		*p*-Value		
	Female	Male	Female	Male	Treat	Sex	Treat × Sex
Body dimensions ^2^ (cm)							
Crown–rump length	39.52 ± 0.43	40.60 ± 0.55	40.33 ± 0.33	39.87 ± 0.44	0.59	0.71	0.09
Thoracic girth	32.26 ± 0.67	32.09 ± 0.83	32.85 ± 0.51	31.75 ± 0.66	0.71	0.26	0.48
Front leg length	29.11 ± 0.94	31.29 ± 1.2	29.90 ± 0.75	29.19 ± 1.00	0.77	0.64	0.15
Hind leg length	33.40 ± 0.42	33.35 ± 0.54	33.43 ± 0.33	33.75 ± 0.45	0.70	0.70	0.67
Organ weight (g) ^2^							
Semitendinosus m	5.26 ± 0.24	5.12 ± 0.3	5.20 ± 0.18	5.21 ± 0.24	0.99	0.85	0.74
Brain	53.73 ± 1.38	55.59 ± 1.64	51.17 ± 1.05	52.25 ± 1.23	0.08	0.22	0.73
Heart	24.37 ± 1.06	27.08 ± 1.26	25.61 ± 0.80	25.12 ± 0.95	0.97	0.36	0.08
Lungs	121.94 ± 8.08	124.19 ± 10.08	105.03 ± 6.19	109.47 ± 7.96	0.07	0.64	0.89
Liver	79.02 ± 4.61	80.64 ± 5.67	75.13 ± 3.52	76.62 ± 4.41	0.43	0.72	0.99
Spleen	5.41 ± 0.42	6.50 ± 0.53	5.32 ± 0.32	5.36 ± 0.42	0.28	0.27	0.23
Right kidney	8.31 ± 0.60	9.93 ± 0.71	9.40 ± 0.46	9.22 ± 0.83	0.54	0.25	0.07
Left kidney	8.45 ± 0.67	9.76 ± 0.82	9.57 ± 0.51	9.14 ± 0.63	0.52	0.65	0.17
Perirenal fat	18.77 ± 1.63	16.11 ± 2.05	17.60 ± 1.25	15.55 ± 1.65	0.61	0.17	0.87
Thymus	12.90 ± 1.28	13.29 ± 1.59	13.00 ± 0.98	13.23 ± 1.25	0.98	0.81	0.95
Thyroid glands	0.86 ± 0.07	0.91 ± 0.09	1.04 ± 0.05	0.96 ± 0.07	0.11	0.72	0.37
Adrenal glands	0.41 ± 0.02	0.43 ± 0.03	0.40 ± 0.02	0.40 ± 0.03	0.40	0.59	0.80
Digestive tract weight (g) ^2^							
Stomach	37.76 ± 2.84	38.76 ± 3.41	34.19 ± 2.16	39.64 ± 2.57	0.53	0.13	0.37
Intestine ^3^	120.05 ± 6.11	128.75 ± 7.55	133.77 ± 4.63	140.17 ± 5.84	0.07	0.20	0.85

^1^ Data are presented as the means (predicted from the fitted statistical model) ± SEM, together with the *p* value for treatment, sex and treatment by sex interaction effects. ^2^ Data were analyzed with the fetal weight as covariate. ^3^ Small and large.

**Table 4 animals-14-00946-t004:** Blood gases, chemistries, hematocrit and hemoglobin of fetuses from orally supplemented with NCG and control (CON) ewes at 140 days of gestation.

	CON		NCG		*p-*Value		
	Female	Male	Female	Male	Treat	Sex	Treat × Sex
Sodium, mmol/L	142.47 ± 0.73 ^a^	143.45 ± 0.79 ^b^	142.80 ± 0.58 ^a^	142.22 ± 0.60 ^a^	0.78	0.89	<0.05
Potassium, mmol/L	4.76 ± 0.13	5.04 ± 0.16	4.72 ± 0.10	5.01 ± 0.13	0.77	0.02	0.99
Calcium, mmol/L	1.45 ± 0.02 ^a^	1.53 ± 0.02 ^b^	1.46 ± 0.01 ^a^	1.46 ± 0.01 ^a^	0.21	0.01	<0.01
Hemoglobin, g/dL	12.01 ± 0.63	12.03 ± 0.75	12.96 ± 0.48	12.36 ± 0.56	0.30	0.51	0.57
pH	7.46 ± 0.01 ^a^	7.39 ± 0.01 ^b^	7.44 ± 0.01 ^a^	7.43 ± 0.01 ^a^	0.78	<0.01	<0.01
PCO_2_, mm Hg	38.95 ± 1.52 ^a^	45.05 ± 1.66 ^b^	40.32 ± 1.22 ^a^	41.09 ± 1.28 ^a,b^	0.73	<0.01	<0.01
PO_2_, mm Hg	22.18 ± 1.85	19.53 ± 2.17	20.23 ± 1.41	20.40 ± 1.61	0.67	0.50	0.34
TCO, mmol/L	29.13 ± 0.64	28.24 ± 0.69	28.82 ± 0.52	28.79 ± 0.54	0.98	0.21	0.15
HCO_3_, mmol/L	27.90 ± 0.63	27.16 ± 0.67	27.58 ± 0.51	27.54 ± 0.53	0.95	0.29	0.25
BE, mmol/L	3.87 ± 0.63	2.44 ± 0.69	3.55 ± 0.50	3.12 ± 0.53	0.93	0.02	0.15
sO_2_, %	40.64 ± 4.89	29.96 ± 5.69	34.16 ± 3.74	35.17 ± 4.21	0.71	0.31	0.13
Hematocrit, % PCV	35.35 ± 1.84	35.39 ± 2.20	38.14 ± 1.39	36.42 ± 1.65	0.29	0.51	0.58

Partial pressure of carbon dioxide (PCO_2_), partial pressure of oxygen (PO_2_), total carbon dioxide (TCO_2_), bicarbonate (HCO_3_), base excess (BE), oxygen saturation (sO_2_) value, packed cell volume (PCV). Data are presented as the means (predicted from the fitted statistical model) ± SEM, together with the *p* value for treatment, sex and treatment by sex interaction effects. Different superscripts depict pairwise differences for the interaction effects that were significant (*p* < 0.05).

**Table 5 animals-14-00946-t005:** Plasma metabolites from supplemented (NCG) and control (CON) ewes at 140 days of gestation.

	CON	NCG	*p*-Value
Urea (mmol/L)	7.61 ± 0.18	7.02 ± 0.16	0.07
NEFA (µmol/L)	860.25 ± 62.57	970.90 ± 55.96	0.21
Total protein (g/L)	76.13 ± 1.69	73.00 ± 1.5	0.19
Globulin (g/L)	34.00 ± 1.31	33.40 ± 1.17	0.74
Calcium (mmol/L)	2.15 ± 0.10	2.21 ± 0.09	0.67
Magnesium (mmol/L)	1.08 ± 0.02	1.04 ± 0.02	0.23
Cholesterol (mmol/L)	1.78 ± 0.10	1.89 ± 0.09	0.41
AST (IU/L)	106.13 ± 4.12	109.00 ± 3.68	0.61
CK (IU/L)	104.88 ± 16.72	72.90 ± 14.96	0.17
GGT (IU/L)	54.63 ± 4.46	54.40 ± 3.99	0.97
GLDH (IU/L)	12.50 ± 2.02	13.80 ± 1.57	0.62
Bilirubin (mmol/L)	8.83 ± 0.29	9.49 ± 0.26	0.11
Triglycerides (mmol/L)	0.36 ± 0.06	0.44 ± 0.05	0.35

Non-esterified fatty acids (NEFA), aspartate aminotransferase (AST), creatine kinase (CK), gamma-glutamyltransferase (GGT) and glutamate dehydrogenase (GLDH).

**Table 6 animals-14-00946-t006:** Placental traits from supplemented (NCG) and control (CON) ewes at 140 days of gestation.

Trait	CON	NCG	*p*-Value
Placental weight (g)	599.4 ± 52.78	590.7 ± 45.71	0.90
Placentome weight (g)	560.6 ± 45.13	548.1 ± 37.76	0.83
Placentome number	89.0 ± 4.13	84.0 ± 3.46	0.37
Mean placentome weight (g)	6.3 ± 0.44	6.5 ± 0.37	0.68
Placental efficiency ^1^	11.8 ± 0.88	11.6 ± 0.67	0.86

^1^ Estimated as a ratio of fetal and placental weight.

## Data Availability

The data presented in this study are available in the study.

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
