# Peer review of "The Effect of N-Carbamylglutamate Supplementation during the Last Third of Gestation on the Growth and Development of Fetuses Born to Nutrient-Restricted Twin-Bearing Ewes"

_animals, 2024, doi:10.3390/ani14060946_

Round 1
Reviewer 1 Report
Comments and Suggestions for Authors
General comments
-Why use the ewes in two groups with an not equal population? (NCG n = 10, Control n = 8).
- How do you know that feeding natural pasture and supplementing with alfalfa hay at 600 g/h/d is a nutrient restriction. What have you compared with the recommendations?
-Introduction: Added previous study use NCG supplementation on fetal growth.
-So few references (n = 31), especially those associated with NCG in ruminant. Please add references to NCG in an introduction and discussion.
Specific comments
L181: “if P ≤ 0.05 and as a trend if 0.05 > P ≤ 0.1” changed to “P<0.05 and as a trend, P<0.10.” as a similar with Tables 3 and 4.
Table 2-5: “P”Changed to “P-value”.
Comments on the Quality of English Language
English language should be improve
Author Response
Comments to Reviewer 1
Thank you very much for taking the time to review this manuscript. Please find the detailed responses below and the corresponding revisions/corrections in track changes in the re-submitted files
- Why use the ewes in two groups with an not equal population? (NCG n = 10, Control n = 8).
Answer: The number of twin-bearing ewes expected at the time of the ultrasound was significantly lower to the expected. This was attributed to a seasonal nutrient restriction in the region, affecting the prolificacy of the animals. Therefore, in order to give more statistical robustness to the data of NCG supplemented ewes, more animals were allocated in this group. Previous studies comparing the effect of NCG supplementation to twin-bearing ewes have worked with n=8 per group (Zhang, H., L. W. Sun, Z. Y. Wang, M. T. Deng, G. M. Zhang, R. H. Guo, T. W. Ma, and F. Wang. "Dietary N-carbamylglutamate and rumen-protected L-arginine supplementation ameliorate fetal growth restriction in undernourished ewes." Journal of Animal Science 94, no. 5 (2016): 2072-2085.), therefore, the authors are confident that the number of animals was sufficient to determine the potential differences between treatments.
- How do you know that feeding natural pasture and supplementing with alfalfa hay at 600 g/h/d is a nutrient restriction. What have you compared with the recommendations?
Answer: According to NRC, twin-bearing ewes at late gestation with a weight of 60 kg require 3.94 Mcal/ME and 173 g of CP to cover their requirements. In our research, nutrient composition of natural pastures (4.23% CP, 1.69 Mcal/kg ME) and alfalfa (13% CP and ME 2.2 Mcal/kg ME, dry matter basis), even considering a complete intake of DM (1.6 kg DM/day), was not sufficient to cover those requirements. Therefore, animals were maintained in a nutrient restricted condition during the whole period of the experiment. We would like to mention that ewes were supplemented with alfalfa hay only from 130 to 140 dga due to severe nutrient restriction resulting from environmental conditions.
- Introduction: Added previous study use NCG supplementation on fetal growth.
Answer: The following references were included in the Introduction as well as in the Discussion section, in order to acknowledge previous reports in the effect of NCG in other ruminants. In addition, new bibliography was included to support key topics, related to our research. The list of the new bibliography included is:
- Refshauge, G.; Brien, F.D.; Hinch, G.N.; Van De Ven, R. Neonatal Lamb Mortality: Factors Associated with the Death of Australian Lambs. Anim. Prod. Sci. 2016, 56.
- Scales, G.H.; Burton, R.N.; Moss, R.A. Lamb Mortality, Birthweight, and Nutrition in Late Pregnancy. New Zeal. J. Agric. Res. 1986, 29.
- Wu, G.; Morris Jr, S.M. Arginine Metabolism: Nitric Oxide and Beyond. Biochem. J. 1998, 336, 1–17.
- Zhang, C.Z.; Sang, D.; Wu, B.S.; Li, S.L.; Zhang, C.H.; Jin, L.; Li, J.X.; Gu, Y.; Ga, N.M.R.; Hua, M.; et al. Effects of Dietary Supplementation with N-Carbamylglutamate on Maternal Endometrium and Fetal Development during Early Pregnancy in Inner Mongolia White Cashmere Goats. Anim. Sci. J. 2022, 93.
- Zhang, H.; Zha, X.; Zhang, B.; Zheng, Y.; Liu, X.; Elsabagh, M.; Ma, Y.; Wang, H.; Shu, G.; Wang, M. Dietary Rumen-Protected L-Arginine or N-Carbamylglutamate Enhances Placental Amino Acid Transport and Suppresses Angiogenesis and Steroid Anabolism in Underfed Pregnant Ewes. Anim. Nutr. 2023, 15.
- Zhang, H.; Liu, X.; Zheng, Y.; Zhang, Y.; Loor, J.J.; Wang, H.; Wang, M. Dietary N-Carbamylglutamate or L-Arginine Improves Fetal Intestinal Amino Acid Profiles during Intrauterine Growth Restriction in Undernourished Ewes. Anim. Nutr. 2022, 8.
- Zhang, H.; Zhao, F.; Nie, H.; Ma, T.; Wang, Z.; Wang, F.; Loor, J.J. Dietary N -Carbamylglutamate and Rumen-Protected l -Arginine Supplementation during Intrauterine Growth Restriction in Undernourished Ewes Improve Fetal Thymus Development and Immune Function. Reprod. Fertil. Dev. 2018, 30.
- Gu, F.; Jiang, L.; Xie, L.; Wang, D.; Zhao, F.; Liu, J. Supplementing N-Carbamoylglutamate in Late Gestation Increases Newborn Calf Weight by Enhanced Placental Expression of MTOR and Angiogenesis Factor Genes in Dairy Cows. Anim. Nutr. 2021, 7.
- Robinson, J.J. Nutritional Requirements of the Pregnant and Lactating Ewe. In Genetics of Reproduction in Sheep; 1985.
- Gootwine, E.; Spencer, T.E.; Bazer, F.W. Litter-Size-Dependent Intrauterine Growth Restriction in Sheep. Animal 2007, 1, 547–564.
- Peine, J.L.; Jia, G.; Van Emon, M.L.; Neville, T.L.; Kirsch, J.D.; Hammer, C.J.; O’Rourke, S.T.; Reynolds, L.P.; Caton, J.S. Effects of Maternal Nutrition and Rumen-Protected Arginine Supplementation on Ewe Performance and Postnatal Lamb Growth and Internal Organ Mass. J. Anim. Sci. 2018, 96.
- Sales, F.; Peralta, O.; Narbona, E.; McCoard, S.; De los Reyes, M.; González-Bulnes, A.; Parraguez, V. Hypoxia and Oxidative Stress Are Associated with Reduced Fetal Growth in Twin and Undernourished Sheep Pregnancies. Animals 2018, 8.
- Zhang, H.; Zhao, F.; Peng, A.; Dong, L.; Wang, M.; Yu, L.; Loor, J.J.; Wang, H. Effects of Dietary l -Arginine and N -Carbamylglutamate Supplementation on Intestinal Integrity, Immune Function, and Oxidative Status in Intrauterine-Growth-Retarded Suckling Lambs. J. Agric. Food Chem. 2018, 66.
- Zhang, H.; Sun, H.; Peng, A.; Guo, S.; Wang, M.; Loor, J.J.; Wang, H. N -Carbamylglutamate and l-Arginine Promote Intestinal Function in Suckling Lambs with Intrauterine Growth Restriction by Regulating Antioxidant Capacity via a Nitric Oxide-Dependent Pathway. Food Funct. 2019, 10.
- Gu, F.F.; Jiang, L.Y.; Wang, D.M.; Zhao, F.Q.; Liu, J.X. Supplementation with N-Carbamoylglutamate during the Transition Period Improves the Function of Neutrophils and Reduces Inflammation and Oxidative Stress in Dairy Cows. J. Dairy Sci. 2022, 105.
- So few references (n = 31), especially those associated with NCG in ruminant. Please add references to NCG in an introduction and discussion.
Answer: number of references were increased as suggested and described in point 3 of Comments to Reviewer 1.
- L181: “if P ≤ 0.05 and as a trend if 0.05 > P ≤ 0.1” changed to “P<0.05 and as a trend, P<0.10.” as a similar with Tables 3 and 4.
Answer: Changes were made as suggested by the reviewer.
- Table 2-5: “P” changed to “P-value”.
Answer: Thank you for pointing this out. Changes were made in all tables as suggested.
- English language should be improved
Answer: The document was reviewed and English improved as suggested.
Reviewer 2 Report
Comments and Suggestions for Authors
Comments to the Authors
The current manuscript provides information maternal body weight, body condition score, blood metabolites, and placental morphology; and fetal weight, organ weights and blood biochemistry in naturally nutrient-restricted grazing twin-bearing ewes supplemented with N-carbamylglutamate from 100 days of gestation to term. The aim of this study was to evaluate the effect of oral N-carbamylglutamate supplementation from 100 days to term in naturally nutrient-restricted grazing twin-bearing ewes, on maternal body weight, and body condition score and blood metabolites, placental morphology, and fetal weight, body composition and blood biochemistry.
The experiment was balanced with a sufficient number of animals per treatment. The authors have done a good job of substantiating their claims with a well-referenced methodology. The manuscript is well illustrated with legible illustrations. This work is a contribution to the difficulty of meeting the nutritional needs of pregnant ewes under difficult environmental conditions. It reinforces the need for future research to identify new strategies to improve lamb survival in such environments worldwide. The manuscript could be considered for publication after addressing the following shortcomings.
A few details:
Line 108-109: Deleted ‘‘prepared’’
Author Response
Comments to Reviewer 2
Thank you very much for taking the time to review this manuscript. Please find the detailed responses below and the corresponding revisions/corrections in track changes in the re-submitted files.
A few details: Line 108-109: Deleted ‘‘prepared’’
Answer: Thank you for pointing this out. The word ‘‘prepared’’ was deleted as suggested.
Reviewer 3 Report
Comments and Suggestions for Authors
General comments:
This study aimed to evaluate the effect of N-carbamylglutamate (NCG) supplementation (60 mg/kg) from day 100 to 140 (term) of pregnancy on several maternal and fetal traits of naturally nutrient-restricted grazing ewes carrying twins. For this purpose, the body weight and BCS of ewes were periodically measured as well 15 maternal plasma metabolites. At day 140 fetus were assessed for morphological measurements, blood gases, hematology and biochemistry. Two oxidative stress biomarkers (TAC and MDA) were evaluated in both ewes and offsprings. The study design was well conceived and in accordance with previous ones reported in literature for this subject. This aspect is also evident in introduction and discussion sections. The statistical analysis is adequate, but some care is required in the presentation of results (tables) regarding what is significant and what is a tendency. The number of animals (twin-bearing ewes) was reported in M&M (n=18), but I suggest to report univocally that the samples are based in these ewes and in 36 fetuses. This is important due to the tendency (0.05 > P ≤ 0.1) for differences between groups observed in several outcomes. As conclusion, the authors suggest specific issues for further studies. This aspect mitigates the negative results of several traits observed in the present study and was well appreciated. Minor changes are suggested to clarify some points.
Specific comments:
L115: It is in a dry matter basis?
L124: Placentomes or the cotyledons (after placentome dissection)?
L226, 255: It is not usual to insert superscript letter for tendencies: what pairs are significant (P<0.05)? What post hoc test was used?
L234. I suggest to move this sentence to discussion section.
L263: I suggest to reorder TAC and MDA.
L230-234: I suggest to start the results section with this finding.
L306: I suggest to add the NCG concentration: (60 mg/kg).
Author Response
Comments to Reviewer 3
Thank you very much for taking the time to review this manuscript. Please find the detailed responses below and the corresponding revisions/corrections in track changes in the re-submitted files.
- L115: It is in a dry matter basis?
Answer: Thank you for pointing this out. The estimation is based on dry mater, therefore it was included in the paragraph.
- L124: Placentomes or the cotyledons (after placentome dissection)?
Answer: In this case we are referring to placentomes, as maternal and fetal components of the placentome were dissected together.
- L226, 255: It is not usual to insert superscript letter for tendencies: what pairs are significant (P<0.05)? What post hoc test was used?
Answer: Thank you for pointing this out. The Reviewer is correct when mentioning that it is not usual to use superscripts for tendencies. They were included, in order to clarify the reader how groups differed when a tendency was observed. As suggested, superscripts were deleted. Table 3 and Table 4 footnotes were modified accordingly.
Post hoc multiple comparisons were conducted using Fisher’s Least Significant Difference (LSD) Test. The following text was included in the reviewed version: “When presented, post-hoc tests were performed using Fisher’s Least Significant Difference (LSD) Test”.
- I suggest to move this sentence to discussion section.
Answer: After reviewing the structure of the discussion section, moving the sentence from results was not possible, in order to maintain a coherent structure of the section. Therefore, the sentence was maintained in the results section.
- L263: I suggest to reorder TAC and MDA.
Answer: Thank you for pointing this out. The sentence was reordered as suggested.
- L230-234: I suggest to start the results section with this finding.
Answer: Thank you for pointing this out. The finding was moved at the beginning of the results as suggested and subtitle numbers were adjusted accordingly.
- L306: I suggest to add the NCG concentration: (60 mg/kg).
Answer: As suggested by the reviewer, the NCG concentration was included.
Reviewer 4 Report
Comments and Suggestions for Authors
In the manuscript submitted for the review the Authors present the effect of oral N-carbamylglutamate administration to twin-bearing ewe from 100 day of gestation to term, kept in naturally nutrient-restricted grazing on different variables from dams, foetuses and placenta. The Authors did not demonstrated a significant influence of N-carbamylglutamate administrated in mid-to-late gestation on maternal and foetal traits important to support lamb survival and growth. However, the study is very interesting and purposeful, especially because it was already shown that N-carbamylglutamate may influence the survival and growth of ewes when administrated at different time. The Authors shed a light on the use of the studied substance and They indicate the need for future studies involving interaction between supplementation and the level of nutritional restriction, timing of intervention, and using appropriate doses.
The introduction briefly highlight the study assumptions and materials and methods are properly and thoroughly described. The results are interesting, accurate and are comprehensively discussed.
As the reviewer I have several comments:
- In line 52, there should be “is” instead of “in”
- In line 100, the abbreviation im should be changed into IM or i.m.
- In line 105, the abbreviation dga should be explained in the text in the place where it appears for the first time. It was explained in the Simple Summary but it should be explained also in the text.
- In line 108, the word prepared in the end of the sentence is not necessary.
- In statistical analysis, it should be mentioned if the conditions for statistical tests are fulfilled. For example, both ML and REML methods require normal distribution of the data in the model parameter and residual term.
- In line 198, there is (P>0.05), the rest of the statistical results contain an exact value (for example P = 0.023). The Authors should decide which type of data presentation to choose and use only one of them throughout the text.
- In line 227 in figure 3 it should be stated that the difference P<0.05 is between NCG and Control. Moreover, the Authors already used the CON abbreviation for control group so it should be used in the graph instead of control.
- In line 367, the abbreviation dga should be used instead day of gestation as previously
- In reference 9, there is a lack of data e.g. name of journal.
In my opinion, the manuscript entitled: The effect of N-carbamylglutamate supplementation during last third of gestation in nutrient restricted twin-bearing ewes on fetal growth is suitable for publication in Animals after minor revision.
Author Response
Comments to Reviewer 4
Thank you very much for taking the time to review this manuscript. Please find the detailed responses below and the corresponding revisions/corrections in track changes in the re-submitted files.
- In line 52, there should be “is” instead of “in”
Answer: Thank you for pointing this out. The word “in” was changed to “is” as suggested.
- In line 100, the abbreviation im should be changed into IM or i.m.
Answer: Thank you for pointing this out. The word “im” was changed to “i.m.” as suggested.
- In line 105, the abbreviation dga should be explained in the text in the place where it appears for the first time. It was explained in the Simple Summary but it should be explained also in the text.
Answer: The line was changed to “…100 days of gestation (dga).” as suggested, to explain in the text the meaning of the abbreviation.
- In line 108, the word prepared in the end of the sentence is not necessary.
Answer: Thank you for pointing this out. The word “prepared” was changed to “i.m.” deleted as indicated.
- In statistical analysis, it should be mentioned if the conditions for statistical tests are fulfilled. For example, both ML and REML methods require normal distribution of the data in the model parameter and residual term.
Answer: The following sentence was added to the statistical analysis description: “All data were assessed for normality and homogeneity and no transformation was necessary.”
- In line 198, there is (P>0.05), the rest of the statistical results contain an exact value (for example P = 0.023). The Authors should decide which type of data presentation to choose and use only one of them throughout the text.
Answer: In this case a general P value (P>0.05) was used as it makes reference to a more than one trait (fetal and organ weights), therefore, the use of the exact P value is not possible. The whole document was checked for similar notations and values were changed in the Maternal Plasma Metabolites section, as recommended by the reviewer.
- In line 227 in figure 3 it should be stated that the difference P<0.05 is between NCG and Control. Moreover, the Authors already used the CON abbreviation for control group so it should be used in the graph instead of control.
Answer: The figure was modified and word “Control” was changed to “CON” as suggested. In addition, the figure description was changed to “*P < 0.05 between NCG and CON.” As suggested.
- In line 367, the abbreviation dga should be used instead day of gestation as previously
Answer: As suggested, the abbreviation was used instead of day of gestation.
- In reference 9, there is a lack of data e.g. name of journal.
Answer: Thank you for pointing this out. The name of the journal “Amino Acids” was included in the reference.